# Omni-Scale CNNs: a simple and effective kernel size configuration for time series classification

**Wensi Tang**[1], **Guodong Long**[1], **Lu Liu**[1,2], **Tianyi Zhou**[3,4], **Michael Blumenstein**[1], **Jing Jiang**[1]

[1]Australian Artificial Intelligence Institute, University of Technology Sydney, [2] Google
[3]University of Washington, Seattle, [4]University of Maryland, College Park
{wensi.tang, lu.liu-10}@student.uts.edu.au,
{guodong.long, michael.blumenstein, jing.jiang}@uts.edu.au, tianyizh@uw.edu

## Abstract

The Receptive Field (RF) size has been one of the most important factors for One Dimensional Convolutional Neural Networks (1D-CNNs) on time series classification tasks. Large efforts have been taken to choose the appropriate size because it has a huge influence on the performance and differs significantly for each dataset. In this paper, we propose an Omni-Scale block (OS-block) for 1D-CNNs, where the kernel sizes are decided by a simple and universal rule. Particularly, it is a set of kernel sizes that can efficiently cover the best RF size across different datasets via consisting of multiple prime numbers according to the length of the time series. The experiment result shows that models with the OS-block can achieve a similar performance as models with the searched optimal RF size and due to the strong optimal RF size capture ability, simple 1D-CNN models with OS-block achieves the state-of-the-art performance on four time series benchmarks, including both univariate and multivariate data from multiple domains. Comprehensive analysis and discussions shed light on why the OS-block can capture optimal RF sizes across different datasets. Code available here [1]

## 1 Introduction

One of the most challenging problems for Time Series Classification (TSC) tasks is how to tell models in what time scales [2] to extract features. Time series (TS) data is a series of data points ordered by time or other meaningful sequences such as frequency. Due to the variety of information sources (e.g., medical sensors, economic indicators, and logs) and record settings (e.g., sampling rate, record length, and bandwidth), TS data is naturally composed of various types of signals on various time scales (Hills et al., 2014; Schäfer, 2015; Dau et al., 2018). Thus, in what time scales can a model "see" from the TS input data has been a key for the performance of TS classification.

Traditional machine learning methods have taken huge efforts to capture important time scales, and the computational resource consumption increase exponentially with the length of TS increase. For example, for shapelet methods (Hills et al., 2014; Lines et al., 2012), whose discriminatory feature is obtained via finding sub-sequences from TS that can be representative of class membership, the time scale capture work is finding the proper sub-sequences length. To obtain the proper length, even for a dataset with length 512, (Hills et al., 2014) has to try 71 different sub-sequence lengths. For other methods, such as (Berndt & Clifford, 1994; Schäfer, 2015; Lucas et al., 2019), despite the time scale capture might be called by different names such as finding warping size or window length. They all need searching works to identify those important time scales. More recent deep learning based methods also showed that they had to pay a lot of attention to this time scale problem. MCNN (Cui et al., 2016) searches the kernel size to find the best RF of a 1D-CNN for every dataset. Tapnet (Zhang et al., 2020) additionally considers the dilation steps. Chen & Shi (2021) also take

---

[1]https://github.com/Wensi-Tang/OS-CNN.
[2]It has different names for different methods. Generally, it refers to the length of the time series subsequence for feature extraction.

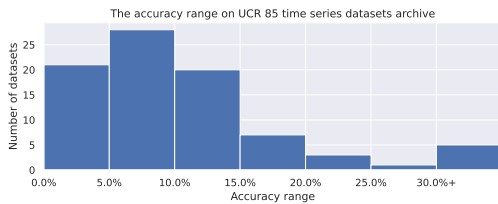 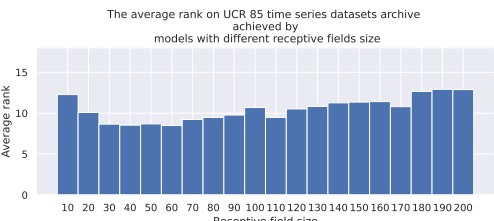

Figure 1: *Left*: A model's accuracy on the UCR 85 datasets changes by tuning the model's receptive field sizes from 10 to 200. *Right*: The average rank results of each receptive field size are pretty similar, which means that no single receptive field size can significantly outperform others on most datasets.

the number of layers into considerations. These are all important factors for the RF of CNNs and the performance for TSC.

Although a number of researchers have searched for the best RF of 1D-CNNs for TSC, there is still no agreed answer to 1) what size of the RF is the best? And 2) how many different RFs should be used? Models need to be equipped with different sizes and different numbers of RFs for a specific dataset. Using the same setup for every dataset can lead to a significant performance drop for some datasets. For example, as shown by the statistics on the University of California Riverside (UCR) 85 "bake off" datasets in Figure 1a, the accuracy of most datasets can have a variance of more than 5% *just* by changing the RF sizes of their model while keeping the rest of the configurations the same. As also shown in Figure 1b, no RF can consistently perform the best over different datasets.

To avoid those complicated and resource-consuming searching work, we propose Omni-Scale block (OS-block), where the kernel choices for 1D-CNNs are automatically set through a simple and universal rule that can cover the RF of all scales. The rule is inspired by Goldbach's conjecture, where any positive even number can be written as the sum of two prime numbers. Therefore, the OS-block uses a set of prime numbers as the kernel sizes except for the last layer whose kernel sizes are 1 and 2. In this way, a 1D-CNN with these kernel sizes can cover the RF of all scales by transforming TS through different combinations of these prime size kernels. What's more, the OS-block is easy to implement to various TS datasets via selecting the maximum prime number according to the length of the TS.

In experiments, we show consistent state-of-the-art performance on four TSC benchmarks. These benchmarks contain datasets from different domains, i.e., healthcare, human activity recognition, speech recognition, and spectrum analysis. Despite the dynamic patterns of these datasets, 1D-CNNs with our OS-block robustly outperform previous baselines with the unified training hyper-parameters for all datasets such as learning rate, batch size, and iteration numbers. We also did a comprehensive study to show our OS-block, the no time scale search solution, always matches the performance with the best RF size for different datasets.

## 2 MOTIVATIONS

Two phenomena of 1D-CNNs inspire the design of the OS-block. In this section, we will introduce the two phenomena with examples in Figure 2 and more discussions can be found in Section 4.6.

Firstly, we found that, although the RF size is important, the 1D-CNNs are not sensitive to the specific kernel size configurations that we take to compose that RF size. An example is given in the right image of the Figure 2

Secondly, the performance of 1D-CNNs is mainly determined by the best RF size it has. To be specific, supposing we have multiple single-RF-size-models which are of similar model size and layer numbers, but each of them has a unique RF size. Let's denote the set of those RF sizes as $\mathbb{S}$. When testing those models on a dataset, we will have a set of accuracy results $\mathbb{A}$. Then, supposing we have a multi-kernel model which has multiple RF sizes[3] and set of those sizes is also $\mathbb{S}$. Then,

---

[3]A detailed discussion about how to calculate RF sizes for 1D-CNNs with multiple kernels in each layer can be found in Section 3.2

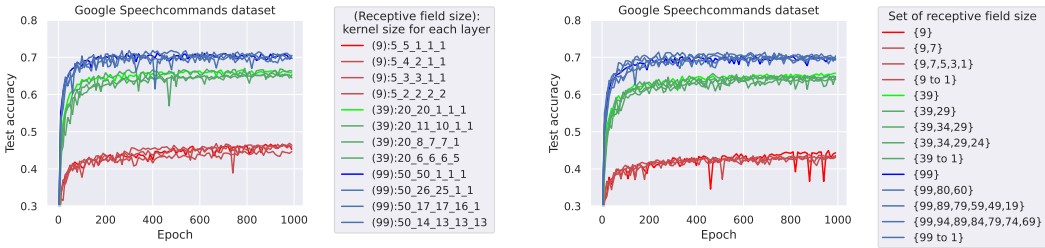

Figure 2: *Left*: The label of each line denotes receptive field size and the kernel configuration of each 1D-CNN. For example, (9):5_5_1_1_1 means the 1D-CNN has five layers and the receptive field size is 9, and from the first layer to the last layer, kernel sizes of each layer are 5, 5, 1, 1, and 1. Lines of similar color are 1D-CNNs with the same receptive field size, and they are also of similar performance. *Right*: Lines with similar colors are models which have the same best receptive field size. For example, all (red/green/blue) lines have the receptive field size (9/39/99), and their performances are similar to the bright (red/green/blue) line which denotes the model only has the receptive field size (9/39/99).

the accuracy of the multiple-RF-sizes-model will be similar to the highest value of $\mathbb{A}$. An example is given in the left image of Figure 2. Specifically, when testing single-RF-size-models on the Google Speechcommands dataset, the model's performance is **positive correlation** with the model's RF size. For example, the light blue line whose set of RF size is $\{99\}$ outperforms the light green line $\{39\}$ and light red line$\{9\}$. For those multiple-RF-sizes-models which has more than one element in their set of RF sizes, their performance are determined by the best (also the largest because of the **positive correlation**) RF size it has. Having more worse (smaller) RF sizes will not have much influence on the performance.

The second phenomenon means that, instead of searching for the best time scales, if the model covers all RF sizes, its performance will be similar to that of a model with the best RF size. However, there are many designs that can cover all RF sizes. Which one should be preferred? Based on the first phenomenon, from the performance perspective, we could choose any design that we want. However, as we will show in Section 3.3, those candidate designs are not of the same characteristics such as the model size or the expandability for long TS data. Therefore, the design of the OS-block that we propose aims at covering all RF sizes in an efficient manner.

## 3 METHOD

The section is organized as follows: Firstly, we give the problem definition in Section 3.1. Then, we will explain how to construct the Omni-scale block (OS-block) which covers all receptive field sizes in Section 3.2. Section 3.3 will explain the reason why OS-block can cover RF of all sizes in an efficient manner. In Section 3.4, we will introduce how to apply the OS-block on TSC tasks.

### 3.1 PROBLEM DEFINITION

TS data is denoted as $\boldsymbol{X} = [\boldsymbol{x}_1, \boldsymbol{x}_2, ..., \boldsymbol{x}_m]$, where $m$ is the number of variates. For univariate TS data, $m = 1$ and for $m > 1$, the TS are multivariate. Each variate is a vector of length $l$. A TS dataset, which has $n$ data and label pairs, can be denoted as: $\mathbb{D} = \{(\boldsymbol{X}^1, y^1), (\boldsymbol{X}^2, y^2), ..., (\boldsymbol{X}^n, y^n)\}$, where $(\boldsymbol{X}^*, y^*)$ denotes the TS data $\boldsymbol{X}^*$ belongs to the class $y^*$. The task of TSC is to predict the class label $y^*$ when given a TS $\boldsymbol{X}^*$.

### 3.2 ARCHITECTURE OF OS-BLOCK

The architecture of the OS-block is shown in Figure 3. It is a three-layer multi-kernel structure, and each kernel does the same padding convolution with input. For the kernel size configuration, we use $\mathbb{P}^{(i)}$ to denote the kernel size set of the $i$-th layer:

$$\mathbb{P}^{(i)} = \begin{cases} \{1, 2, 3, 5, ..., p_k\} & , i \in \{1, 2\} \\ \{1, 2\} & , i = 3 \end{cases} \tag{1}$$

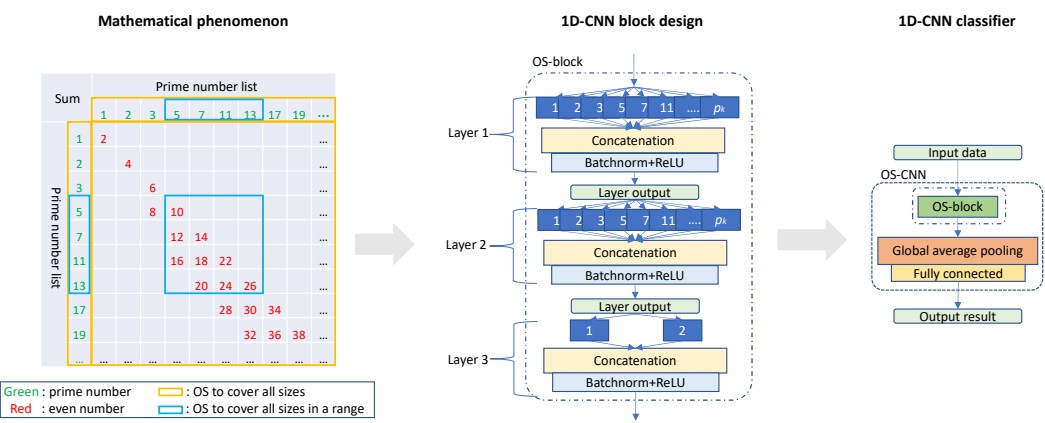

Figure 3: The *left* image shows that every even number from 2 to 38 can be composed via two prime numbers from 1 to 19. This phenomenon can be extended to all even numbers. Based on this phenomenon, with the OS-block structure in the *middle* image, we could cover all receptive field sizes. Specifically, the first two layers have prime-sized kernels from 1 to $p_k$. Thus, the two layers can cover all even number receptive field sizes. With kernels of sizes 1 and 2 in the third layer, we could cover all integer receptive field sizes in a range via selecting the value $p_k$. The OS-block is easy to be applied on time series classification tasks. A simple classifier with the OS-block, namely OS-CNN, is given in the *right* image, which achieves a series of SOTA performances.

Where $\{1, 2, 3, 5, 7, ..., p_k\}$ is a set of prime numbers from 1 to $p_k$. The value of $p_k$ is the smallest prime number that can cover all sizes of RF in a range. Here, the range that we mentioned is all meaningful scales. For example, since the TS length is $l$, we don't need to cover RFs that are larger than $l$ or smaller than 1. Therefore, the $p_k$ is the smallest prime number that can cover the RF size from 1 to $l$. If we have prior knowledge, such as that we know there are cycles in the TS, or we know the length range of the hidden representative pattern. We could change the RF size range of the OS-block by simply changing the prime number list. An example is given in the left image in Figure 3, which uses the prime number list in the blue block to cover the RF size range from 10 to 26.

**RF sizes of the OS-block**: The RF is defined as the size of the region in the input that produces the feature. Because each layer of the OS-block has more than one convolution kernel, there will be several different paths from the input signal to the final output feature (Araujo et al., 2019; Luo et al., 2016), and each path will have a RF size. For the 3-layer OS-block, which has no pooling layer and the stride size is 1, the set of RF sizes $\mathbb{S}$ is the set of RF size of all paths, and it can be described as:

$$\mathbb{S} = \{p^{(1)} + p^{(2)} + p^{(3)} - 2 \mid p^{(i)} \in \mathbb{P}^{(i)}, i \in \{1, 2, 3\}\}. \tag{2}$$

For the reasons that $\mathbb{P}^{(i)}$ are prime number list when $i \in \{1, 2\}$, the set $\{p^{(1)} + p^{(2)} | p^{(i)} \in \mathbb{P}^{(i)}, i \in \{1, 2\}\}$ is the set of all even numbers $\mathbb{E}$.[4] Thus, we have

$$\mathbb{S} = \{e + p^{(3)} - 2 \mid p^{(3)} \in \mathbb{P}^{(3)}, e \in \mathbb{E}\}. \tag{3}$$

With Equation 3 and Equation 1, we have

$$\mathbb{S} = \{e|e \in \mathbb{E}\} \cup \{e - 1|e \in \mathbb{E}\} \equiv \mathbb{N}^+. \tag{4}$$

Where $\mathbb{N}^+$ is the set of all integer numbers in the range. Specifically, the $\mathbb{S} \equiv \mathbb{N}^+$ is because a real number must be an odd number or an even number, while $\mathbb{E}$ is the even number set, $\{e - 1|e \in \mathbb{E}\}$ is

---

[4]This is according to Goldbach's conjecture. Specifically, the conjecture states that any positive even number can be composed of two prime numbers. For example, $8 = 5 + 3$, $12 = 7 + 5$, and more examples can be found in the left image of Figure 3. Despite that the conjecture is yet unproven in theory, but its correctness has been validated up to $4 \times 10^{14}$ (Richstein, 2001), which is larger than the length of all available TS data.

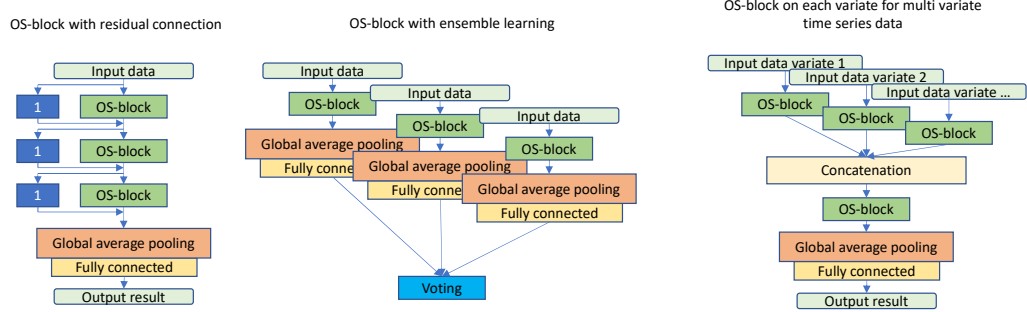

Figure 4: Examples of using OS-block with other deep learning structures.

the odd number set. Therefore, with the proper selection of $p_k$, we could cover any integer RF size in a range. It should be noticed that, there might be many options to cover all RF sizes, we use the Godlach's conjecture to make sure that we could all scales.

### 3.3 OS-BLOCK COVER ALL SCALES IN AN EFFICIENT MANNER

From the model size perspective, using prime numbers is more efficient than using even numbers or odd numbers. To be specific, to cover receptive fields up to size r, the model size complexity of using prime size kernels is $O(r^2/log(r))$. On the other hand, no matter we use even number pairs or odd number pairs, the model size complexity is $O(r^2)$. We also empirically show the advantage of our model on efficiency in the following table and this table has been added to Appendix A.7:

### 3.4 HOW TO APPLY OS-BLOCK ON TSC TASKS

Firstly, the OS-block could take both univariate and multivariate TS data by adjusting the input channel the same as the variate number of input TS data. A simple example classifier with OS-block, namely OS-CNN, is given in Figure 3. The OS-CNN is composed of an OS-block with one global average pooling layer as the dimensional reduction module and one fully connected layer as the classification module. Other than OS-CNN, the OS-block is flexible and easy to extend. Specifically, convolution layers of OS-block can be calculated parallelly. Thus, each layer can be viewed as one convolutional layer with zero masks. Therefore, both the multi-kernel layers or the OS-block itself are easy to extend with more complicated structures (such as dilation (Oord et al., 2016), attention or transformer (Shen et al., 2018a), and bottleneck) that are normally used in 1D-CNN for performance gain. In Figure 4, we give three examples which uses OS-block with other structures.

## 4 EXPERIMENT

### 4.1 BENCHMARKS

We evaluate OS-block on 4 TSC benchmarks which include, in total, 159 datasets. The details of each benchmark is as follows:

- **Magnetoencephalography recording for Temporal Lobe Epilepsy diagnosis (MEG-TLE)** dataset (Gu et al., 2020): The Magnetoencephalography dataset was recorded from epilepsy patients and was introduced to classify two subtypes (simple and complex) of temporal Lobe Epilepsy. The dataset contains 2877 recordings which were obtained at the sampling frequency 1200 Hz. Each recording is approximately 2 sec. Therefore the length is about 2400.

- **University of East Anglia (UEA) 30 archive** (Bagnall et al., 2018): This formulation of the archive was a collaborative effort between researchers at the University of East Anglia and the University of California, Riverside. It is an archive of 30 multivariate TS datasets

| Individual dataset benchmark | | | | |
|---|---|---|---|---|
| Dataset | Method | Accuracy(%) | F1-score | # parameters |
| | CNN (Gu et al., 2020) | 83.2 | 82.3 | 3.8M |
| | PF (Gu et al., 2020) | 82.6 | 68.2 | - |
| MEG-TLE | SVM (Gu et al., 2020) | 55.2 | 85.2 | - |
| (Gu et al., 2020) | MSAM (Gu et al., 2020) | 83.6 | 83.4 | 2.3M |
| | Rocket (Dempster et al., 2020) | 87.7 | 89.9 | - |
| | **OS-CNN (Ours)** | **91.3** | **91.6** | **235k** |

| Multivariate dataset archive benchmark | | | | | |
|---|---|---|---|---|---|
| Archive | Method | Baseline wins | **OS-CNN (Ours)** wins | Tie | Average Rank |
| | DTW-1NND(norm) (Zhang et al., 2020) | 7 | **23** | 0 | 5.68 |
| | DTW-1NN-I(norm) (Zhang et al., 2020) | 5 | **24** | 1 | 6.70 |
| | ED-1NN(norm) (Zhang et al., 2020) | 5 | **25** | 0 | 7.45 |
| | DTW-1NND (Zhang et al., 2020) | 7 | **23** | 0 | 5.28 |
| UEA 30 archive | DTW-1NN-I (Zhang et al., 2020) | 7 | **22** | 1 | 6.07 |
| (Bagnall et al., 2018) | ED-1NN (Zhang et al., 2020) | 5 | **25** | 0 | 7.12 |
| | WEASEL+MUSE (Schäfer & Leser, 2017) | 10 | **19** | 1 | 4.15 |
| | MLSTM-FCN (Karim et al., 2019) | 7 | **23** | 0 | 5.62 |
| | TapNet (Zhang et al., 2020) | 9 | **20** | 1 | 3.80 |
| | **OS-CNN (Ours)** | - | - | - | **3.13** |

| Univariate dataset archives benchmarks | | | | | |
|---|---|---|---|---|---|
| Archive | Method | Baseline wins | **OS-CNN (Ours)** wins | Tie | Average rank |
| | PF (Lucas et al., 2019) | 13 | **67** | 5 | 6.57 |
| | ResNet (Wang et al., 2017) | 19 | **61** | 5 | 5.41 |
| | STC (Hameurlain et al., 2017) | 27 | **56** | 2 | 5.05 |
| UCR 85 archive | InceptionTime (Ismail Fawaz et al., 2019) | 34 | **42** | 9 | 4.05 |
| (Chen et al., 2015) | ROCKET (Dempster et al., 2020) | 33 | **44** | 8 | 3.64 |
| | HIVE-COTE (Lines et al., 2016) | 34 | **43** | 8 | 3.99 |
| | TS-CHIEF (Shifaz et al., 2020) | **42** | 39 | 4 | 3.68 |
| | **OS-CNN (Ours)** | - | - | - | **3.59** |
| | ResNet (Wang et al., 2017) | 19 | **83** | 26 | 3.21 |
| UCR 128 archive | InceptionTime (Ismail Fawaz et al., 2019) | 30 | **59** | 39 | 2.41 |
| (Dau et al., 2018) | ROCKET (Dempster et al., 2020) | 43 | **62** | 23 | 2.36 |
| | **OS-CNN (Ours)** | - | - | - | **2.02** |

Table 1: Performance comparison on 4 time series classification benchmarks

from various domains such as motion detection, physiological data, audio spectra classification. Besides domains, those datasets also have various characteristics. For instance, among those datasets, the class number various from 2 to 39, the length of each dataset various from 8 to 17,894, and the number of variates various from 2 to 963.

- **University of California, Riverside (UCR) 85 archive** (Chen et al., 2015): This is an archive of 85 univariate TS datasets from various domains such as speech reorganizations, health monitoring, and spectrum analysis. What's more, those datasets also have different characteristics. For instance, among those datasets, the class number varies from 2 to 60, the length of each dataset varies from 24 to 2709. The number of training data varies from 16 to 8,926.

- **University of California, Riverside (UCR) 128 archive** (Dau et al., 2018): This is an archive of 128 univariate TS datasets. It is the updated version of the UCR 85 archive. However, the new archive cannot be viewed as a replacement for the former because they have different characteristics. For example, for the UCR 85 archive, all TS data within a single dataset are of the same length, but that is not the same for the UCR 128 archive. Besides that, in general, the added data in the UCR 128 archive, their default test set is bigger than the train set to reflect real-world scenarios.

## 4.2 EVALUATION CRITERIA

For all benchmarks, we follow the standard settings from previous literature. Specifically, for the MEG-TLE dataset, following Multi-Head Self-Attention Model (MSAM) (Gu et al., 2020), models are evaluated by test accuracy and f1 score. Besides using recommended metrics of each benchmark, we also compare the model size of OS-block with other deep learning methods. For UEA 30, UCR 85 archives, and UCR 128 archives, following the evaluation advice from the archive (Dau et al., 2018; Bagnall et al., 2018), count of wins, and critical difference diagrams (cd-diagram) (Dau et al., 2018) were selected as the evaluation method. Due to the page limitation, we list the average rank

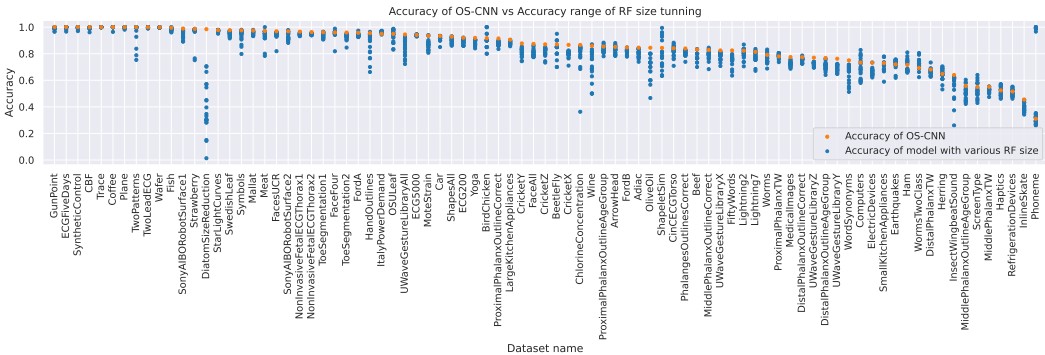

Figure 5: Classification accuracies for OS-CNN vs. accuracies from 20 1D-CNNs with receptive field size. As we can see, for most of the dataset, the orange points (accuracy of OS-CNN) are near the top of blue points. More analysis for this comparison can be found in Appendix A.1

in the result table because it is the main criteria of the cd-diagram. The full cd-diagram results are listed in Appendix A.3.

## 4.3 EXPERIMENT SETUP

For the MEG-TLE dataset, they were normalized by z-normalization (Chen et al., 2015). For the other archives, we take the raw dataset without processing for datasets in those archives already normalized with z-normalization.

Following the setup of (Wang et al., 2017), we use the learning rate of 0.001, batch size of 16, and Adam (Kingma & Ba, 2014) optimizer. The baselines are chosen from the top seven methods from the leaderboard [5] of each benchmark. For UCR archives, we ensemble five OS-CNNs, which stacks two OS-blocks with residential connections followed by the baseline IncpetionTime (Ismail Fawaz et al., 2019). We use PyTorch [6] to implement our method and run our experiments on Nvidia Titan XP.

## 4.4 STATE-OF-THE-ART PERFORMANCE ON BENCHMARKS

We show consistent state-of-the-art performance on four benchmarks as in Table 1. As we can see, for the MIT-TLE dataset, OS-CNN outperforms baselines in a ten times smaller model size. For all dataset archives, the OS-block achieves the best average rank, which means that, in general, the OS-block design can achieve better performance.

## 4.5 OS-BLOCK CAN CAPTURE THE BEST TIME SCALE

To demonstrate that the OS-block can capture the best time scale, we build 20 FCN models with different RF sizes (from 10 to 200 with step 10), and compare their performance with OS-CNN on the UCR 85 archive. Specifically, the FCN (Wang et al., 2017) is selected as the backbone model for it has a similar structure as OS-CNN.

To obtain FCN with various RF sizes, we change the kernel size of each layer proportionally. To be specific, the kernel sizes of the original three layer FCN are 8, 5, and 3, and the RF size is 14. To obtain the RF size 30, we will set kernel sizes of each layer as 16,10, and 6. To control variables, when the kernel size increases, we will reduce the channel number to keep the model size constant. This will not influence the conclusion. To check that, in Appendix A.2, we also provide the static result comparison between OS-CNN and FCNs with the fixed channel number.

---

[5]http://www.timeseriesclassification.com/results.php

[6]https://pytorch.org/

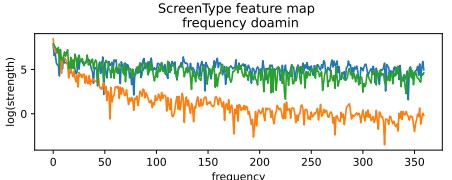 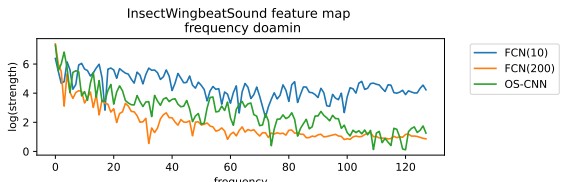

Figure 6: The class activation map of OS-CNN is similar to that of the model which has a better performance. For the ScreenType dataset, FCN(10) outperforms FCN(200), and the class activation map of OS-CNN (green) is similar to FCN(10)(blue). For the InsectWingbeatSound dataset, the class activation map is similar to FCN(200) for FCN(200) outperforms FCN(10).

Due to the page limitation, the full result can be found in the supplementary material. And in Figure 5, a simple result comparison is given, and we could see that for most of the datasets, OS-CNN can achieve a similar result as models with the best time scale.

### 4.6 DISCUSSION ABOUT BEST TIME SCALE CAPTURE ABILITY

The result in Figure 5 empirically verifies two phenomena that we mentioned in Section 2 with multiple datasets from multiple domains. Firstly, the OS-block covers all scales. Therefore, besides the important size, it also covers many redundant sizes, but those redundancies will not pull down the performance. Secondly, OS-block composes the RF size via the prime design while the FCN uses a different design. It means that the performances of 1D-CNNs are determined mainly by the RF size instead of the kernel configuration to compose that.

### 4.7 CASE STUDY FOR THE BEST TIME SCALE CAPTURE ABILITY

To further demonstrate the RF size capture ability, we will give a case study that compares the class activation map (Zhou et al., 2016) of OS-CNN with that of models with the best RF size. We select the ScreenType and InsectWingbeatSound datasets for the case study. They were selected because they are of the largest and the smallest accuracy difference calculated by the accuracy of FCN with RF size 10 (FCN(10)) minus accuracy of FCN with RF size 200 (FCN(200)). Specifically, it can be seen as, among UCR 85 datasets, the ScreenType is the dataset which the FCN(10) outperform FCN(200) most, and InsectWingbeatSound is the dataset which the FCN(200) outperforms FCN(10) most. We visualize the class activation map of the first instance in the two datasets, and the results are shown in Figure 6. As we can see in Figure 6, the class activation map of the OS-block is similar to that of the model with the best RF size.

## 5 RELATED WORKS

A TS data is a series of data points. TSC aims at labeling unseen TS data via a model trained by labeled data (Dau et al., 2018; Chen et al., 2015). One well-known challenge for TSC is telling the model in what time scale to extract features (Hills et al., 2014; Schäfer, 2015; Berndt & Clifford, 1994). This is because TS data is naturally composed of multiple signals on different scales (Hills et al., 2014; Schäfer, 2015; Dau et al., 2018) but, without prior knowledge, it is hard to find those scales directly.

The success of deep learning encourages researchers to explore its application on TS data (Längkvist et al., 2014; Fawaz et al., 2019; Dong et al., 2021). The Recurrent Neural Network (RNN) is designed for temporal sequence. In general, it does not need extra hyper-parameters to identify information extraction scales. However, RNN is rarely applied on TS classification (Fawaz et al., 2019). There are many reasons for this situation. One widely accepted reason is that when faced with long TS data, RNN models suffer from vanishing gradient and exploding gradient (Pascanu et al., 2013; Fawaz et al., 2019; Bengio et al., 1994).

Nowadays, the most popular deep-learning method for TSC is 1D-CNN. However, for 1D-CNNs, the feature extraction scale is still a problem. For example, there is an unresolved challenge with

kernel size selection where there exists different approaches but non consensus on which is best. To date, the selection of feature extraction scales for 1D-CNN is regarded as a hyper-parameter selection problem e.g., (Cui et al., 2016) uses a grid search to find kernel sizes, while the following methods tune it empirically (Zheng et al., 2014; Wang et al., 2017; Rajpurkar et al., 2017; Serrà et al., 2018; Ismail Fawaz et al., 2019; Kashiparekh et al., 2019).

**Dilated convolution** (Oord et al., 2016) is widely adopted in 1D-CNN to improve generalization ability for TS tasks (Oord et al., 2016; Zhang et al., 2020; Li et al., 2021). It takes a lower sampling frequency than the raw signal input thus can be viewed as a structure-based low bandpass filter. Compared with the OS-block, the dilated convolution also needs prior knowledge or searching work to set the dilation size which will determine the threshold to filter out redundant information from TS data.

**Inception structure** (Szegedy et al., 2015) is widely used in 1D-CNN for TSC tasks (Ismail Fawaz et al., 2019; Kashiparekh et al., 2019; Chen & Shi, 2021; Dong et al., 2021). The design of the multi-kernel structure of OS-block is inspired from the inception structure (Szegedy et al., 2015). Compared with existing works, the OS-block has two differences. Firstly, the OS-block does not need to assign weight to important scales via complicated methods such as pre-train (Kashiparekh et al., 2019), attention (Chen & Shi, 2021; Shen et al., 2018b), or a series of modifications such as bias removal and bottleneck for convolutions (Ismail Fawaz et al., 2019). Secondly, OS-block does not need to search for candidate scales. Specifically, those methods can only assign weight to a limited number of scales. Thus, they still need searching works to answer a series of questions. For example, Which sequence, such as geometric or arithmetic, should be preferred? What's the largest length to stop? How do they select the depth of the neural network? And how do they select the common difference or ratio for their sequence?

**Adaptive receptive field** (Han et al., 2018; Tabernik et al., 2020; Xiong et al., 2020; Pintea et al., 2021; Liu et al., 2021; Tomen et al., 2021; Dong et al., 2021), has been proposed to learn the optimal kernel sizes during the training stage. Generally, it can be viewed as learning a weight mask on kernels to control the receptive field size. The weight of the mask can be learned during the training step. On the other hand, OS-block learns the linkage between kernels and uses kernels of different sizes to compose different receptive field sizes. In principle, the adaptive receptive field can be used on the time series classification tasks. It improves the performance by enabling 1D-CNNs to have the best receptive field size. But the OS-block targets at covering all sizes of receptive filed sizes and assign large weight on important sizes.

Mathematically, OS-block is a very general technique and can be extended to time series vision tasks by using the prime size design on the time dimension. This is because the video classification task and time series classification task share the same challenge (Xie et al., 2018; Bian et al., 2017; Tan et al., 2021; Liu et al., 2020; Li et al., 2020), which is the same region of interest might of different time scales for different data. Thus, using the kernel of various sizes will increase the probability to catch proper scales. However, in this paper, we mainly target the classic 1D time series classification, which is an active research area with many open problems (Fawaz et al., 2019; Zhang et al., 2020; Dempster et al., 2020) unsolven.

## 6 CONCLUSION

The paper presents a simple 1D-CNN block, namely OS-block. It does not need any feature extraction scale tuning and can achieve a similar performance as models with the best feature extraction scales. The key idea is using prime number design to cover all RF sizes in an efficient manner. We conduct experiments to demonstrate that the OS-block can robustly capture the best time scale on datasets from multiple domains. Due to the strong scale capture ability, it achieves a series SOTA performance on multiple TSC benchmarks. Besides that, the OS-CNN results reveal two characteristics of 1D-CNN models, which will benefit the development of the domain. In the future, we could extend our work in the following aspects. Firstly, other than the prime kernel size design, there might be a more efficient design to cover all RF sizes. Secondly, the OS-block can work with existing deep neural structures to achieve better performance, but there might be unique structures or variants of those existing structures that are more suitable for the OS-block. Besides that, characteristics of OS-block are empirically analyzed via the way there must be a theoretical explanation of the characteristics.

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

# A APPENDIX

## A.1 STATISTIC OF THE COMPARISON (FIX MODEL SIZE)

More statistic results of the result in Figure 5 is shown in Figure 7, Figure 8 and Figure 9

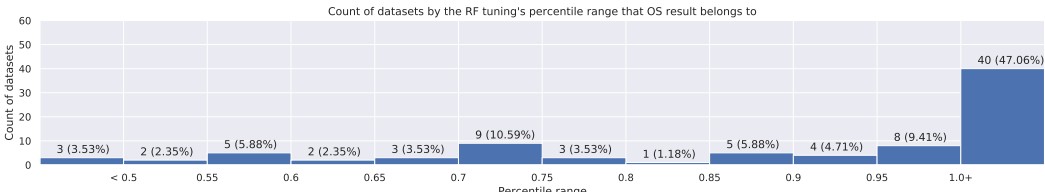

Figure 7: The histogram statics the count of datasets by which percentile range of the blue line that the orange point belongs to. Specifically, we could see that for more than 56% datasets (8+40 out of 85 datasets), the result of OS-block is larger than 0.95 percentile. This means that, for an unknown dataset, using OS-block will have more than 56% chance to achieve a better result than grid search from 20 candidate scales. When seeing the count of the number larger than 0.5 percentile, we could see that, for an unknown dataset, using OS-block will have more than 96% chance to achieve a better result than selecting a random scale.

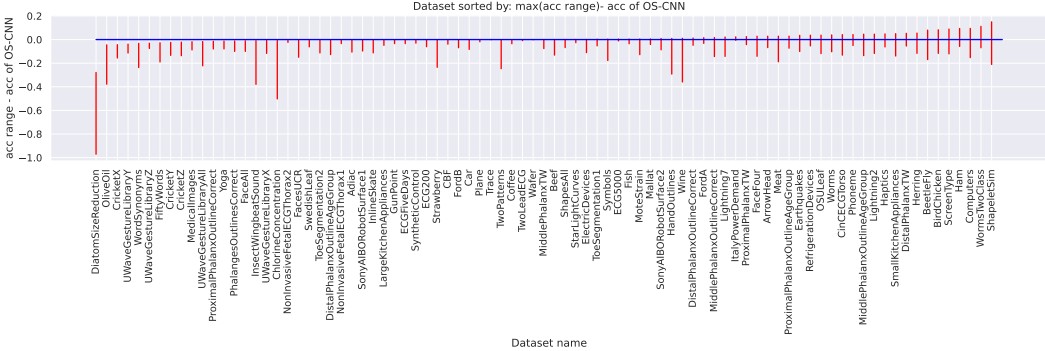

Figure 8: The red line is the accuracy range obtained via subtracting the accuracy of OS-CNN from the accuracy range of FCN with various kernels. We sorted those datasets in ascending order. We could see that for most of the datasets, the highest value of the FCN accuracy range is lower than the accuracy of OS-CNN. Which supports the OS-block has the ability to capture the best scales.

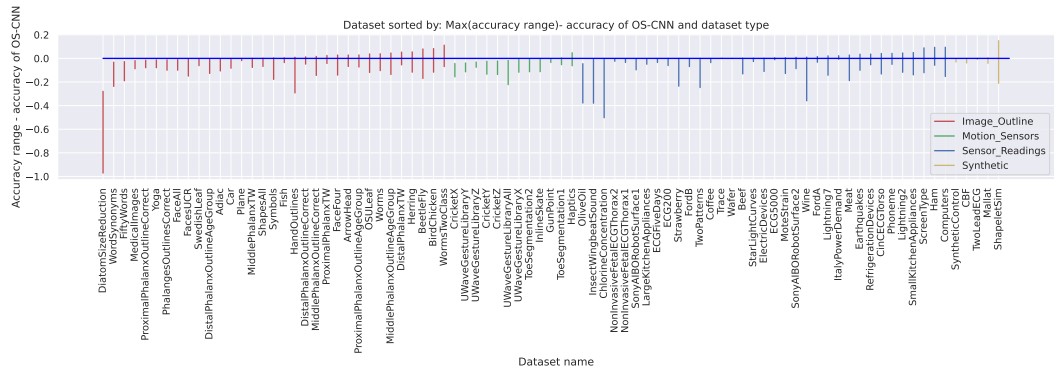

Figure 9: Sort datasets by dataset type and max accuracy range - accuracy of the OS-CNN. We could see that the best scale capture ability keeps the consistency cross different dataset types.

## A.2 STATISTIC OF THE COMPARISON (FIX CHANNEL NUMBER)

When we keep the number of channels constant in FCN, the statistic result will be as this. The OS-CNN still achieves similar performance as the model with the best scales.

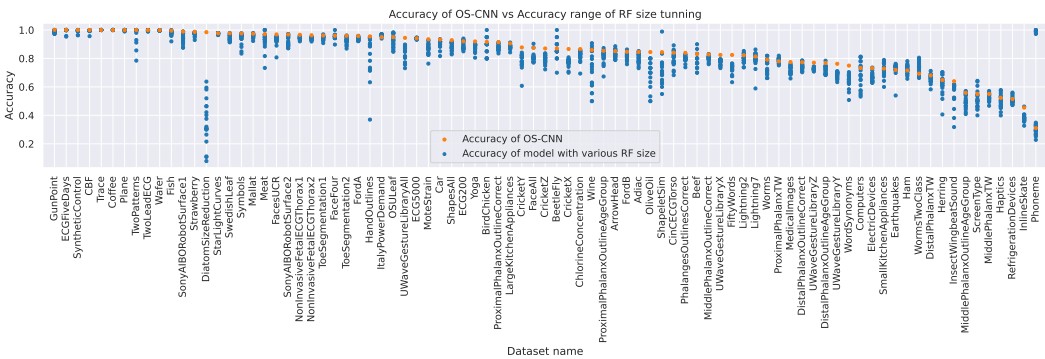

Figure 10: Same static metric as that of Figure 5 and Figure 7

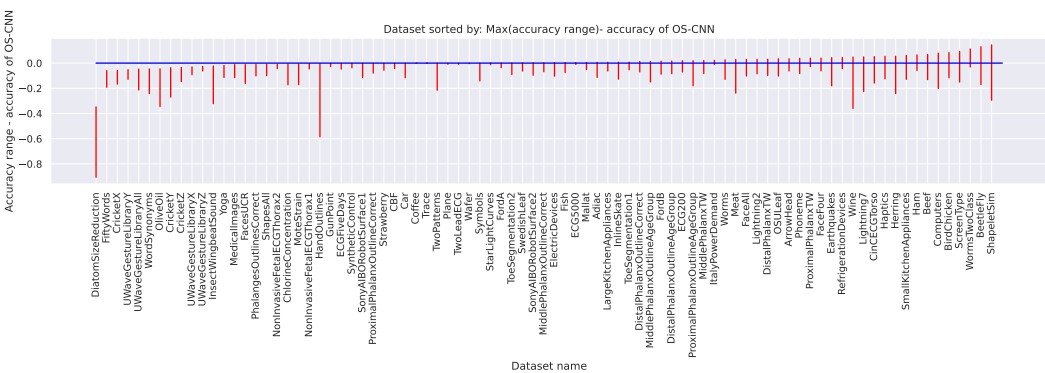

Figure 11: Same static metric as that of Figure 8

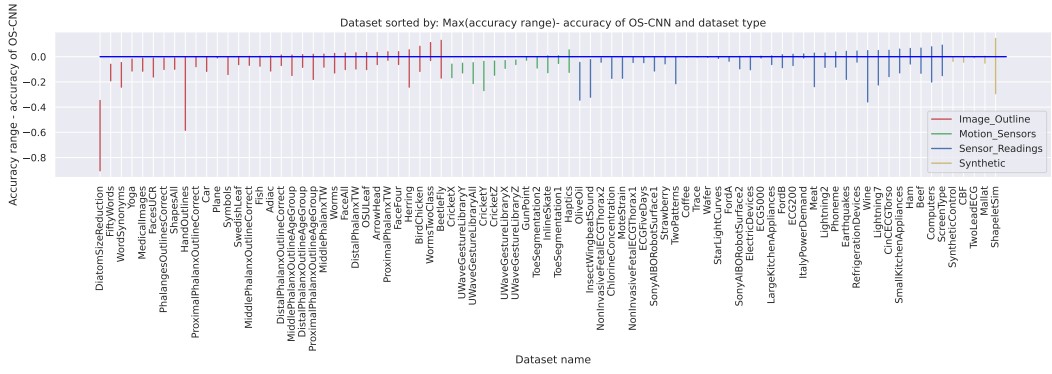

Figure 12: Same static metric as that of Figure 9

### A.3 THE CD-DIAGRAM RESULT

The critical difference diagram shows the average rank of each method with Wilcoxon-Holm post-hoc analysis between each series.

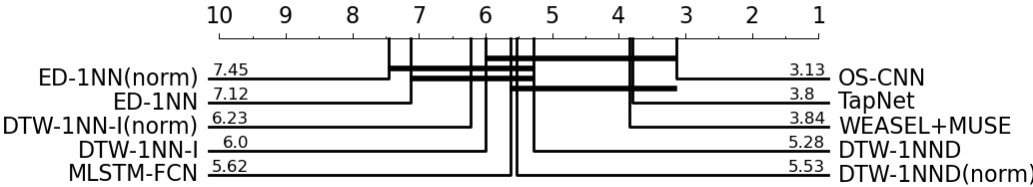

Figure 13: SOTA for UEA 30 multivariate dataset archive

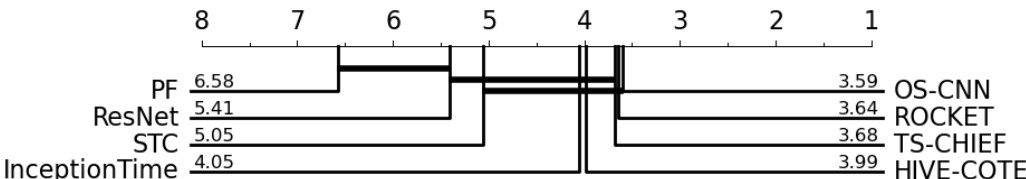

Figure 14: SOTA on the UCR 85 datasets

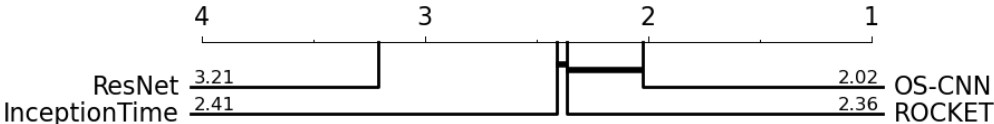

Figure 15: SOTA on the UCR 128 datasets

### A.4 EXAMPLES OF THE TWO PHENOMENA

In the Figure 2, the Google speechcommands dataset is selected as the dataset to show the example. This is because, for this dataset, the relationship between performance and receptive field size is proportional (As it is shown in Figure 16). Thus, it is easy to control variables. What's more, in Figure 18 and Figure 17, we show those two phenomena with more train and test split.

### A.5 EXTEND OS-BLOCK WITH OTHER STRUCTURES

Layers in the OS-block and the OS-block itself are easy to extend with other complicated structures. Figure 19 gives an explanation about the how to view the multi-kernel layers in OS-block as a single layer, and gives an example that how to combine the layer with dilation. The Figure 4 shows that how to view the OS-block as a layer, and gives another two examples rather than OS-CNN.

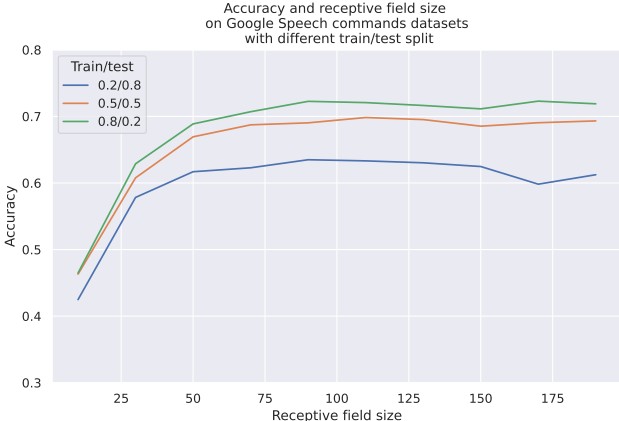

Figure 16: The relationship between performance and receptive field size are proportional

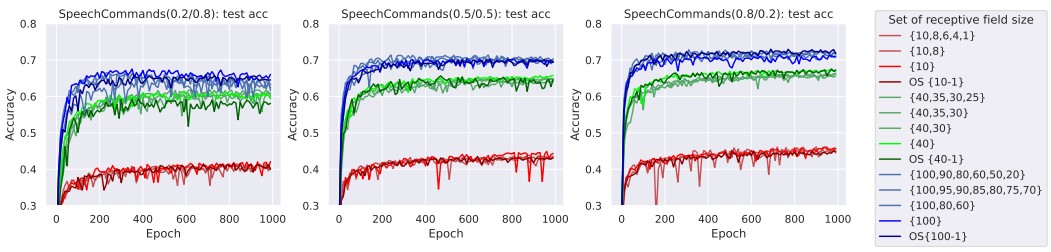

Figure 17: Lines in the figure are models with different sets of receptive field sizes. Lines with similar colors are models which have the same best receptive field size. We could see that the best receptive field size mainly dominates the performance in the set of receptive fieldsizes.

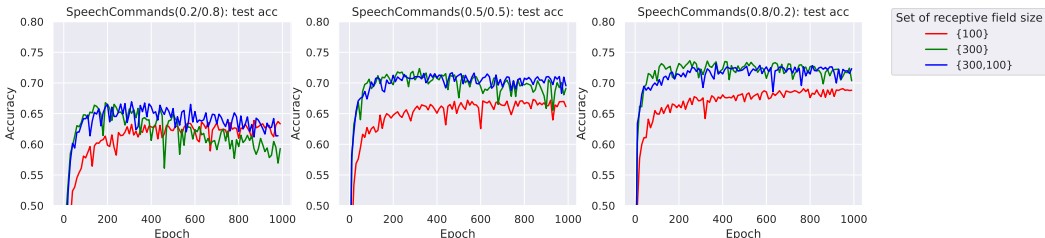

Figure 18: The label of each line denotes the kernel configuration of each 1D-CNN. For example, 5_5_1_1_1 means the 1D-CNN has five layers, and from the first layer to the last layer, kernel sizes of each layer are 5, 5, 1, 1, and 1. Lines of similar color are 1D-CNNs with the same receptive field size, and they are also of similar performance.

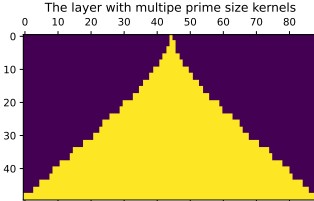 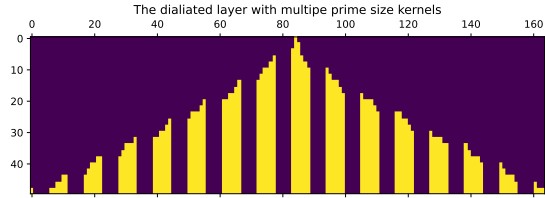

Figure 19: Purple color in those images are the zero mask and yellow denotes the location where has the ability to hold weight. *Left*: Convolution layers in of OS-block can be calculated parallelly, thus, each layer can be viewed as one convolutional layer with zero masks.(s) *Right*: layers in the OS-block can work with the dilation design

A.6   EXPERIMENT RESULT OF OS-BLOCK WITH OTHER STRUCTURES

The Figure 20 and Figure 21 show that applied OS-block with residual connection, ensemble, and multi-channel architectures (individually or together) could further improve the performance. The evaluation was on both UCR 85 and UEA 30 archives which contain datasets from different domains such as electrical devices analysis, Spectrum analysis, traffic analysis, EEG analysis.

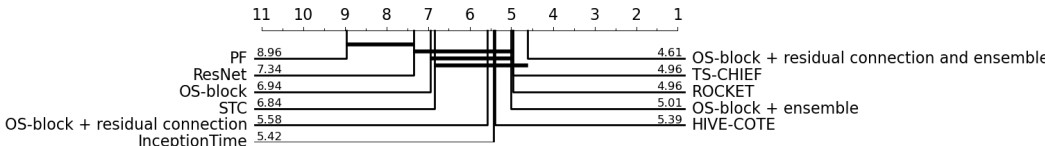

Figure 20: Using the OS-block with residual connection and ensemble (individually or together) could increase the performance

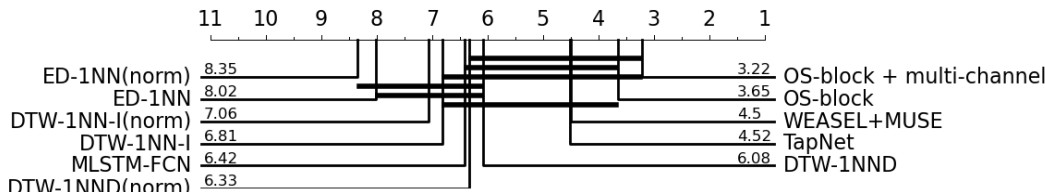

Figure 21: Using the multi-channel architecture with OS-block could improve the performance

A.7   COMPARE THE OS-BLOCK WITH OTHER DESIGNS

Mathematically, finding the optimal kernel configuration is challenging, for it is a constrained combinatorial optimization searching for the best configuration among an exponential number of candidates. Our contribution is a simple and effective model design that does not need to solve the complex optimization problems and achieves state-of-the-art performance on several benchmarks.

From the model size perspective, using prime numbers is more efficient than using even numbers or odd numbers. To be specific, to cover RF of range r, the model size complexity of using prime size kernels is $O(r^2/log(r))$. On the other hand, no matter we use even number pairs or odd number pairs, kernel sizes in each layer, the model size complexity of using the sequence is $O(r^2)$. As Table 2 shows, compared with using odd number pairs or even numbers pairs prime numbers can achieve similar performance in a smaller model size.

| **Number of parameters** | | | | |
|---|---|---|---|---|
| Channel number | RF range | Prime numbers (Ours) | odd numbers | even numbers |
| 16 | 1 to 45 | 304k | 507k | 491k |
| 32 | 1 to 45 | 1,203 k | 2,009k | 1,948k |

| **Accuracy** | | | | |
|---|---|---|---|---|
| Channel number | RF range | Prime numbers (Ours) | odd numbers | even numbers |
| 16 | 1 to 45 | 0.7524 | 0.7687 | 0.7561 |
| 32 | 1 to 45 | 0.7845 | 0.7783 | 0.7725 |

Table 2: Model size and performance comparison on Google SpeechCommands dataset

