# OpenReview forum: "Omni-Scale CNNs: a simple and effective kernel size configuration for time series classification"
_ICLR.cc/2022/Conference — ICLR 2022 Poster_

### Official Review · Reviewer_aFAf · 2021-10-29

**Correctness:** 3
**Technical Novelty And Significance:** 3
**Empirical Novelty And Significance:** 3
**Recommendation:** 6
**Confidence:** 3

**Details Of Ethics Concerns:**

I dont think the missed paper are left out on purpose; thus i see no ethical concerns.

**Main Review:**

Strengths

+ Elegant solution to design an architecture to cover all RFs
+ Good results

Weaknesses

- No related work on adaptive RFs is mentioned at all. See my detailed comments below.
- The paper is sometimes difficult to understand. See my detailed comments below.


All these following citations about adaptive receptive fields in CNNs are missed:

- Han et al. Optimizing filter size in convolutional neural networks for facial action unit recognition. In
Proceedings of the IEEE Conference on Computer Vision and Pattern Recognition (CVPR), 2018

- Tabernik et al. Spatially-adaptive filter units for compact and efficient deep neural networks. International Journal of Computer Vision (IJCV), 2020.

- Zhitong Xiong et al. Variational context-deformable convnets for indoor scene parsing. In Proceedings of the IEEE/CVF Conference on Computer Vision and Pattern Recognition (CVPR), June 2020.

- Pintea et al. Resolution learning in deep convolutional networks using scale-space theory in Transactions on Image Processing (TIP), 2021

- Tomen et al. Deep Continuous Networks in International Conference on Machine Learning (ICML), 2021.

Adaptive receptive fields are quite related to this paper; and I expect them to be included and their relation to this ICLR submission explained and motivated. In addition I would like to see the paper state if these aproaches are, or are not, applicable here.


Detailed comments:

Fig 1: I like the figure. Yet, a "win" is based on the very best score, and thus quite sensitive to small changes; it could be that a method is only slightly worse in which case it would not "win". It would be a stronger plot to also show the average accuracy (or the average rank) for each Receptive field size on all datasets.

Page 2: "Firsty, we find that".. I do not understand what is meant here.

Page 3: It would be good to remind the reader what the term "OS block" means, to make the method section more modular and better readable as a stand-alone text.

Page 3: Section 3.2: At this point in the text I'm confused how these kernel sizes are used in a "multi-kernel structure" as this is not explained.

Fig 3: "medium" = "middle"

Page 4: Section 3.2: I'm still confused here about "because each layer of the OS-block has more than one convolution kernel"; how does that work? Fig 3 seems to hint as something, but is not sufficiently exlained.

Eq 2: I do not understand why a specific RF size is now coupled to it's specific position in the layers (ie: I do not understand why Godlbach's conjecture is needed to uniquely define unique RF size paths through the layers).

Section 3.3. I do not understand what is meant by "For example, suppose we use 2 as the common ratio to cover RF size 15".

Fig 5: There is no need to sort the datasets alphabetically. Please sort the datasets on OS-CNN accuracy: it will probably make the graph much easier to understand. (indeed, in appendix A there seem to be such plots, but I do not understand the diffences between all those plotted variants; I would just sort fig 5 directly on the accuracy of OS-CNN).

Fig 6: the graphs are small, and difficult to see the differences.



**Summary Of The Paper:**


The paper is on the receptive field of CNNs for 1D time series classification. It proposes an elegant decomposition of receptive fields based on the Goldbach conjecture that any number can be represented by a sum of primes. The paper thus puts RFs with prime numbers in CNN layers, and by having multiple layers, these RFs will sum, and thus allowing the proposed network to cover all RF sizes. Experiments on several datasets shows that the method gives good results.

**Summary Of The Review:**

I like the elegant decomposition to cover all RFs. Yet, much related work is missed, and clarity can be improved.

These issues seem fixable.

---

> ### Author Response · Authors · 2021-11-16
> **Response to Reviewer aFAf**
>
> Thanks very much for your constructive comments! Here are our responses and if you have further questions feel free to let us know!
> ```
> Q1. difference between adaptive receptive fields CNNs and OS-block.
> ```
> ***A1:***
>
> In the related work of the new revision, we have added the paragraph below to discuss the differences between the OS-block and adaptive receptive field.
>
> "Adaptive receptive field [1~5] has been proposed to learn the optimal kernel sizes during the training of 2D CNNs. It can be viewed as learning a weight mask on kernels to control the receptive field size. The weight of the mask can be learned during the training step. On the other hand, OS-block applies a manual designed simple rule to choose kernel sizes and efficiently cover a more comprehensive receptive field. In principle, the adaptive receptive field can be used on the time series classification tasks. However, they need additional algorithm designs and training."
>
>
> [1] Han et al. Optimizing filter size in convolutional neural networks for facial action unit recognition. (CVPR), 2018
>
> [2] Tabernik et al. Spatially-adaptive filter units for compact and efficient deep neural networks. (IJCV), 2020.
>
> [3] Zhitong Xiong et al. Variational context-deformable convnets for indoor scene parsing.  (CVPR), June 2020.
>
> [4] Pintea et al. Resolution learning in deep convolutional networks using scale-space theory. (TIP), 2021
>
> [5] Tomen et al. Deep Continuous Networks. (ICML), 2021.
>
> ```
> Q2. Fig 1, It would be a stronger plot to also show the average accuracy (or the average rank) for each Receptive field size on all datasets.
> ```
> ***A2:***
>
> Thanks for the advice, we have replaced the count of win results with the average rank results, which show a similar tendency as the “win” criteria.
>
> ```
> Q3.  Page 2: "Firstly, we find that".. I do not understand what is meant here.
> ```
> ***A3:***
>
> This paragraph aims at showing that the performance of 1D-CNN mainly depends on its receptive field size with almost no regard to the choices of the kernel sizes that can cover the receptive field size (though different kernel size configurations result in different model sizes). This is validated in the left image in Figure2, where a group of lines with similar colors have similar performance because they cover the same receptive field size (with different kernel size configurations).
>
> ```
> Q4. Page 3: It would be good to remind the reader what the term "OS block" means, to make the method section more modular and better readable as a stand-alone text.
> ```
> ***A4:***
>
> In the revision, we have added this change to Section 3 and highlighted it with red color.
>
>
> ```
> Q5. Fig 3: "medium" = "middle"
> ```
> ***A5:***
> Thanks for your advice! We have fixed this typo.
>
> ```
> Q6. Page 3: Section 3.2: At this point in the text I'm confused how these kernel sizes are used in a "multi-kernel structure" as this is not explained.
> Q7. Page 4: Section 3.2: I'm still confused here about "because each layer of the OS-block has more than one convolution kernel"; how does that work?
> ```
> ***A6 and A7:***
>
> There are multiple kernels in each layer and every kernel is applied to the same input and produces an output representation. Then, all output representations will be concatenated along the ***channel dimension***. The concatenated representations will be passed to a bach-norm layer and a ReLU layer. Then, the output will be the input of the next layer.
>
> ```
> Q 8. Eq 2: I do not understand why a specific RF size is now coupled to it's specific position in the layers (ie: I do not understand why Godlbach's conjecture is needed to uniquely define unique RF size paths through the layers).
> ```
> ***A8:***
>
> A specific RF size is not coupled to a specific position in the layers. In Equation 2, the $i$ of $p^{(i)}$ is not the position, $p^{(i)}$ denotes a prime number (the kernel size) in the i-th layer.
>
> Godlach’s conjecture is not used to define the path. As explained in our reply to Q7, we use Godlach’s conjecture to make sure that we could get all odd-size receptive fields after the first two layers.
>
> ```
> Q 9. Section 3.3. I do not understand what is meant by "For example, suppose we use 2 as the common ratio to cover RF size 15".
> ```
> ***A9:***
>
> We improved the statement in the revision. It aims at explaining why to use a set of prime numbers instead of a series of odd numbers or even numbers.
>
>
> ```
> Q 10. Fig 5: Please sort the datasets on OS-CNN accuracy
> ```
> ***A10:***
> We have revised Fig. 5 and sorted them by accuracy as you suggested.
>
>
> ```
> Q 11. Fig 6: the graphs are small, and difficult to see the differences.
> ```
> ***A11:***
> We have increased the size of Fig. 6 in the revision and made it more clear.

---

> > ### Comment · Reviewer_aFAf · 2021-11-28
> > **Thanks**
> >
> > Thanks for this response. Given this response, and the other reviews, I still stand behind my positive verdict.

---

### Official Review · Reviewer_1qMR · 2021-10-31

**Correctness:** 3
**Technical Novelty And Significance:** 2
**Empirical Novelty And Significance:** 2
**Recommendation:** 6
**Confidence:** 4

**Main Review:**

While the proposed method yields a simple and universal rule for kernel size tuning, and has shown to perform as best as the state-of-the-art methods in the literature, there are some major points that are neglected in the current version of the manuscript:

1) The motivation and justification of using the Goldbach’s conjecture is missing. According to the conjecture, any even number such as "e" can be written as the sum of two prime numbers such as "p1" and "p2". Authors have used this decomposition and prime numbers p1 and p2 for the kernel size of their OS-block. However, any even number "e" can also be written as the sum of many other pairs of even numbers such as "e1" and ''e2" or odd numbers such as "o1" and ''o2". Why not using these (e1, e2) pairs or (o1, o2) pairs for the kernel size of the OS-block? What advantage does (p1, p2) have over (e1, e2) or (o1, o2)?

2) As shown in Figure 2, model's performance has a positive correlation with the single RF size. Furthermore, authors have claimed in the beginning of Section 2 that the performance of 1D-CNN is not sensitive to the specific kernel size configuration that leads to a specific RF size. If that is the case, then using the Goldbach’s conjecture makes even less sense since any other decomposition of the even number "e" could perform the same according to this claim.

**Summary Of The Paper:**

This paper presents an Omni-Scale block (OS-block) to efficiently determine the kernel sizes for 1D-CNNs. The proposed method achieves similar performance as models with optimized RF sizes.

**Summary Of The Review:**

The main novelty and contribution of the paper lies in using the Goldback's conjecture to design the OS-block for an efficient kernel size design. However, this key factors suffer from the above two points and the manuscript needs to be revised to address these issue. Authors can either theoretically show the benefit of prime decomposition or experimentally verify the efficacy of the prime decomposition of the even number "e" over other trivial decomposition of "e" such as (e1, e2) or (o1, o2).

---

> ### Author Response · Authors · 2021-11-16
> **Response to Reviewer 1qMR**
>
> Thank you very much for your question! Here is our response to your concerns. We are happy to answer any other questions that you still have!
>
> ```
> Q1. Authors can either theoretically show the benefit of prime decomposition or experimentally verify the efficacy of the prime decomposition of the even number "e" over other trivial decomposition of "e"  such as (e1, e2) or (o1, o2).
> ```
> ***A1:***
>
> From the ***model size*** perspective, using prime numbers is more efficient than using even numbers or odd numbers. To be specific, to cover receptive fields up to size r,  the ***model size complexity*** of using prime size kernels is ***$O(r^{2}/log(r))$***. On the other hand, no matter we use even number pairs or odd number pairs, the ***model size complexity*** is ***$O(r^{2})$***. We also empirically show the advantage of our model on efficiency in the following table and this table ***has been added*** to Appendix A 7 in our revised version:
>
> The table below shows that compared with using odd number pairs or even numbers pairs, prime number design only requires a ***smaller*** model size to achieve ***comparable*** performance.
>
>
>
>  ***Number of parameters***
>
> | Channel number   |  RF range |    Prime numbers (***Ours***)   |    odd numbers     |   even numbers |
> | :-:           |:-:  | :-:   | :-:   |:-: |
> |          16           |  1 to  45     |            ***304k***                      |         507k              |         491k           |
> |             32             |  1 to  45     |             ***1,203 k***                 |       2,009k             |       1,948k          |
>
>
>  ***Accuracies***
>
> | Channel number   |  RF range |    Prime numbers (***Ours***)   |    odd numbers     |   even numbers |
> | :-:           |:-:  | :-:   | :-:   |:-: |
> |          16           |  1 to  45     |            0.7524                      |        0.7687               |        0.7561          |
> |             32             |  1 to  45     |            0.7845                 |      0.7783              |       0.7725          |

---

> > ### Comment · Reviewer_1qMR · 2021-11-25
> > **Respond to author's comment**
> >
> > Thank you for adding these clarifying points to my comment; however, the accuracy results in the above table seem a bit odd. If none of the prime numbers (p1, p2) is equal to 2, then both p1 and p2 should be odd numbers. Therefore, the pair (p1, p2) would be an special case of a pair of odd numbers (o1, o2) and the accuracy reported for odd numbers should be greater than that of the (p1, p2) pair. But, in the above table, the accuracy 0.7783 reported for the odd numbers is less than the accuracy 0.7845 reported for the prime numbers. Would you please provide the value of the prime numbers p1 and p2 used to calculate the accuracy of 0.7845 in the last row of the table?

---

> > > ### Author Response · Authors · 2021-11-28
> > > **The difference is caused by randomness during training**
> > >
> > > Thanks for your kind reply!  Here is our response. We'd like to answer any questions that you still have!
> > > ```
> > > Q1: Would you please provide the value of the prime numbers p1 and p2 used to calculate the accuracy of 0.7845.
> > > ```
> > > ***A1:***
> > >
> > > The set of prime numbers we use is: 1, ***2***, 3, 5, 7, 11, 13, 17, 19, 23.
> > >
> > > The set of odd numbers we use is: 1, 3, 5, 7, ***9***, 11, 13, ***15***, 17, 19 ,***21***, 23.
> > >
> > > ```
> > > Q2 The reported prime result (0.7845) should be lower than the odd result (0.7783)
> > > ```
> > > ***A2:***
> > >
> > > - According to the observation in Figure 2, i.e., the performance of 1D-CNN mainly depends on the best receptive field size it can cover, **the prime result should be similar to the odd result** because they **cover all receptive field sizes**.
> > >
> > > - The small difference in their performance pointed out by you is a result of randomness (random initialization, random mini-batches, random seeds, etc.) in neural nets training. To verify this, we run the experiments again and the result for ***prime is (0.7820)*** and the result for  ***odd is (0.7870)***, i.e., the prime result is slightly lower than the odd result.
> > >
> > > - We would like to run the experiments for more random trials and report the mean and variance to demonstrate the performance similarity. However, one experiment on this dataset requires days to complete on our GPU(NVIDIA TITAN V) machines, so we try to evaluate the difference of prime results and odd results on a smaller dataset by running ***20 random trials***. To avoid the Texas sharpshooter fallacy, we select the first dataset (Adiac dataset) of the UCR archive. The results are reported below.
> > >
> > >
> > >
> > >  ***Accuracies: same channel number***
> > >
> > > Receptive field range: 1 to 45
> > >
> > > Number of (#) channel = 32
> > >
> > > | Designs   |    Max  |    Min     |   Mean | STD | # parameters|
> > > | :-:           |:-:  | :-:   | :-:   |:-: |:-: |
> > > |         ***Prime (ours)***         |        0.852           |        0.765         |0.826|   0.018|  ***1.2M***|
> > > |         Odd                               |        0.849           |        0.777         |0.822|   0.018|         2.0M|
> > > |         Even                              |        0.852            |        0.767         |0.824|    0.021|      1.8M|
> > >
> > > - As shown in the top part of the table, when fixing the ***same number of channels***, the prime design results in a ***smaller model size*** than that of the odd and even designs. At the same time, they have similar mean accuracy and small std. These results indicate that our prime design can result in a smaller model with similar accuracy as the other two designs.
> > >
> > >
> > > To quantitatively evaluate whether the performance of prime design and odd design is similar when they have the same number of channels (hence smaller model for the prime design), we run 20 random trials for each design and then performed two sample T-test for the mean of their performance as two independent variables (Ttest_ind) and two-sample Kolmogorov-Smirnov test (Kstest). The results are reported below and it shows that the two designs have similar performance while prime design results in a smaller model.
> > >
> > >
> > > | Test  |   Null hypothesis  |  Statistic |    P-value    |
> > > | :-:  |:-:  | :-:   | :-:   |
> > > |Ttest_ind   | two-sided   |   0.652 |   0.518 |
> > > |Kstest   |  two-sided  |   0.2  |   0.832 |

---

### Official Review · Reviewer_rAQQ · 2021-11-03

**Correctness:** 4
**Technical Novelty And Significance:** 3
**Empirical Novelty And Significance:** 3
**Recommendation:** 8
**Confidence:** 4

**Main Review:**

The strengths of the papers are as follow :

1- The paper is well structured and easy to follow

2- Tables and figures are very clear.

3- The authors have conducted extensive experiments to support the claim. Moreover, the results are discussed and compared in a consistent manner.

4- The originality of the paper is related to the introduction of a concept from the theory of numbers, namely Goldbach conjecture allowing to cover the receptive fields at different scales

5- The claim is validated on univariate and multivariate time series considering  different tasks.

The weaknesses of the paper :

1- The proposed receptive field module could have been studied for time series vision tasks, namely video classification for instance

2- The work lacks of theoretical study. The latter is very important to study the stability of the representations. Moreover, Goldbach conjecture is not well motivated and analyzed.

3-  The proposed module could be studied under different neural network architectures to assess its genericity, including different tasks and domain.


**Summary Of The Paper:**

The paper proposes an original trick to tune the receptive field kernels for 1D ConvNets to tackle time series problems.

**Summary Of The Review:**

Following the aforementioned consideration, l recommend to accept the paper "6: marginally above the acceptance threshold"

---

> ### Author Response · Authors · 2021-11-16
> **Response to Reviewer rAQQ**
>
> Thanks very much for your constructive comments and questions. Our responses to raised questions and concerns are as follows.
> ```
> Q1 The proposed receptive field module could have been studied for time series vision tasks, namely video classification for instance
> ```
>
>
> ***A1:***
>
> In principle, OS-block is a general technique and can be straightforwardly extended to time series vision tasks by applying our prime size design to the time dimension. However, for the applications in this paper, we mainly focus on 1D time series classification, which is an active research area with many open problems [1,2,3] unsolven. Hence, experiments of video tasks are out of the major scope of this paper. We have added a section to the new version discussing the details of applying OS-block to video classification tasks and the relationship between OS-CNN and [4,5,6]. Those changes will be highlighted in red.
>
> [1] Dempster, A., Schmidt, D. F., & Webb, G. I. (2021, August). Minirocket: A very fast (almost) deterministic transform for time series classification. In Proceedings of the 27th ACM SIGKDD Conference on Knowledge Discovery & Data Mining (pp. 248-257).
>
> [2] Zhang, X., Gao, Y., Lin, J., & Lu, C. T. (2020, April). Tapnet: Multivariate time series classification with attentional prototypical network. In Proceedings of the AAAI Conference on Artificial Intelligence (Vol. 34, No. 04, pp. 6845-6852).
>
> [3] Fawaz, H. I., Forestier, G., Weber, J., Idoumghar, L., & Muller, P. A. (2019). Deep learning for time series classification: a review. Data mining and knowledge discovery, 33(4), 917-963.
>
> [4] Xie, S., Sun, C., Huang, J., Tu, Z., & Murphy, K. (2018). Rethinking spatiotemporal feature learning: Speed-accuracy trade-offs in video classification. In Proceedings of the European conference on computer vision (ECCV) (pp. 305-321).
>
> [5] Bian, Y., Gan, C., Liu, X., Li, F., Long, X., Li, Y., ... & Lin, Y. (2017). Revisiting the effectiveness of off-the-shelf temporal modeling approaches for large-scale video classification. arXiv preprint arXiv:1708.03805.
>
> [6] Li, X., Wang, Y., Zhou, Z., & Qiao, Y. (2020). Smallbignet: Integrating core and contextual views for video classification. In Proceedings of the IEEE/CVF Conference on Computer Vision and Pattern Recognition (pp. 1092-1101).
>
>
> ```
> Q2 The work lacks of theoretical study.  The latter is very important to study the stability of the representations. Moreover, Goldbach conjecture is not well motivated and analyzed.
> ```
> ***A2:***
>
> Mathematically, finding the optimal kernel configuration is challenging because it is a constrained combinatorial optimization searching for the best configuration among an exponential number of candidates. The major goal of this paper is to develop a simple and effective model design that does not need to solve the complex optimization problem but can outperform most commonly adopted configurations and achieve state-of-the-art performance on widely used benchmarks.
>
> The proposed Omni-scale block aims at covering all receptive field sizes. Godlach’s conjecture guarantees that we could cover all odd-size receptive fields after the first two layers. Therefore, with additional kernels of sizes 1 and 2, we can cover all receptive field sizes.
>
> In the revision, we have added the motivation and analysis of using Godlach’s conjecture in section 3 and highlighted it with red color.
>
> ```
> Q3 The proposed module could be studied under different neural network architectures to assess its genericity, including different tasks and domains.
> ```
> ***A3:***
>
> As in section 4.2, we evaluated our method under different tasks and domains. In the new revision, we have added the experiments for different neural architectures in Appendix A.6. Here is a summary:
>
> **Different neural network architectures** (section 3.4):
> OS-block with residual connection, ensemble, and multi-channel architectures (individually or together) could further improve the performance.
>
> **Tasks** (section 4.1):
> Experiments on both univariate time series classification (85 tasks from the UCR archive) and multivariate time series classification (30 tasks from UEA archive) tasks.
>
> **Domains** (section 4.1):
> The UCR and UEA archives have datasets from different domains such as electrical devices analysis, Spectrum analysis, traffic analysis, EEG analysis, etc.
>
>
> **Domains** (section 4.1):
> The UCR and UEA archives have datasets from different domains such as electrical devices analysis, Spectrum analysis, traffic analysis, EEG analysis.

---

### Decision · Program_Chairs · 2022-01-20

**Decision:**

Accept (Poster)

**Comment:**

The paper presents a simple and effective solution to tune the receptive field of CNNs for 1D time series classification. The reviewers think the idea is original and elegant but would appreciate more theoretical insights into the solution.